# Effects of repetitive transcranial magnetic stimulation over the contralesional dorsal premotor cortex on upper limb function in severe ischaemic stroke: study protocol for a randomised controlled trial

Wenjun Dai ![ORCID],[1] Xi Yang,[2] Canhuan Liu,[2] Hongyuan Ding,[3] Chuan Guo,[1] Yi Zhu ![ORCID],[1] Manyu Dong,[2] Yilun Qian,[2] Lu Fang,[1] Tong Wang,[1] Ying Shen[1]

WD, XY, CL, HD and CG contributed equally.

WD, XY, CL, HD and CG are joint first authors.

For numbered affiliations see end of article.

**Correspondence to**
Dr Ying Shen;
shenying@njmu.edu.cn and
Dr Tong Wang;
wangtong60621@163.com

## ABSTRACT

**Introduction** Repetitive transcranial magnetic stimulation (rTMS) is an evidence-based treatment widely recommended to promote hand motor recovery after ischaemic stroke. However, the therapeutic efficacy of rTMS over the motor cortex in stroke patients is currently restricted and heterogeneous. This study aimed to determine whether excitatory rTMS over the contralesional dorsal premotor cortex (cPMd) facilitates the functional recovery of the upper limbs during the postacute stage of severe ischaemic stroke.

**Methods and analysis** This study will be conducted as a single-blind, controlled, randomised study, in which 44 patients with poststroke hemiplegia with a course of disease ranging from 1 week to 3 months and Fugl-Meyer upper limb score ≤22 will be enrolled. The study participants will be randomly assigned to groups A (n=22) and B (n=22). The two groups are based on routine rehabilitation training and drug treatment; group A will be treated with low-frequency (1 Hz) rTMS over the contralesional primary motor cortex (cM1), and group B will be treated with high-frequency (10 Hz) rTMS over cPMd. For 2 weeks, rTMS will be administered once a day, 5 days a week. The primary outcome is the Fugl-Meyer assessment of the upper limb. The secondary outcomes include the Arm Subscore of the Motricity Index, Hong Kong edition of Functional Test for the Hemiplegic Upper Extremity, Modified Barthel Index and Modified Ashworth Scale score of the paralysed pectoralis major and biceps brachii. Furthermore, data of diffusion tensor imaging and functional MRI will be collected. These outcomes will be assessed before and after the completion of the intervention.

**Ethics and dissemination** This study has been approved by the Ethics Committee of the First Affiliated Hospital of Nanjing Medical University (2020 SR-266). The findings of this study will be spread through networks of scientists, professionals and the general public as well as peer-reviewed scientific papers and presentations at pertinent conferences.

**Trial registration number** ChiCTR2000038049

## STRENGTHS AND LIMITATIONS OF THIS STUDY

⇒ This will be a single-blind, controlled, randomised clinical study to compare high-frequency repetitive transcranial magnetic stimulation (rTMS) over contralesional dorsal premotor cortex (cPMd) and low-frequency rTMS over contralesional primary motor cortex (cM1) in severe ischaemic stroke.

⇒ We will use neuronavigation to position the site of cPMd and cM1 stimulation.

⇒ Additionally, various outcome measures, including clinical function and functional MRI, will be included to investigate the therapeutic effects of rTMS and its underlying neuroplasticity mechanisms.

⇒ Because of low-frequency rTMS to the cM1 ranked as an A-level recommendation for improving motor function after stroke, no blank control group undergoing sham transcranial magnetic stimulation (TMS) will be set up.

⇒ Given the nature of TMS interventions, blinding operators is not feasible.

## INTRODUCTION

Ischaemic stroke has emerged as a critical public health problem as the main contributor to significant long-term impairment.[1 2] Among them, the age group of 40–70 years is the primary high-risk age range for stroke, which has relative homogeneity, stability and representativeness.[3] Movement disorders after ischaemic stroke not only reduce the activities of daily living (ADL) and affect the health of patients but also place a heavy strain on the society as a whole.[2 4] Hypofunction of the upper limbs is one of the most common types of dyskinesia following ischaemic stroke, which seriously affects the patient's ADL and hampers rehabilitation progression.[5 6] In ischaemic stroke rehabilitation, improving

the upper extremity functional capacity remains a challenging issue.

The effectiveness of repetitive transcranial magnetic stimulation (rTMS) as ischaemic stroke rehabilitation therapy has been thoroughly studied.[6 7] This technique promotes brain plasticity by providing continuous magnetic pulses to the brain.[8] After stroke, neuroplasticity-induced cortical reconfiguration is necessary for the restoration of motor performance.[9] According to widespread consensus, low-frequency (≤1 Hz) rTMS (LF-rTMS) can reduce neuronal activity and cortical excitability, whereas high-frequency (≥5 Hz) rTMS (HF-rTMS) has the opposite effect.[10] The interhemispheric inhibition (IHI) model is the foundation of the current rTMS intervention method to support the improvement of upper extremity performance in stroke patients.[11] It indicates that decreased excitability of the lesioned primary motor cortex (M1) region after stroke results from 'double obstacles', including the ipsilateral areas and excessive inhibition in the contralateral areas.[12] Therefore, lowering IHI by minimising the excitation of the contralesional side will improve recovery.[13] The IHI model currently serves as the foundation for recommendations regarding the application of rTMS in stroke recovery, with low-frequency rTMS to the contralesional M1 (cM1) ranked as an A-level recommendation for improving motor function.[10] Because the heterogeneity of patients is not taken into account, the clinical efficacy of this rTMS intervention strategy for poststroke motor dysfunction is limited.[10] However, for patients with severe stroke with low structural retention, using the vicariation model may be preferred, which posits that the activity in the remaining network locations compensates for the loss of certain capabilities in damaged areas.[14]

In patients with severe ischaemic stroke, the dorsal premotor cortex (PMd) of the contralesional hemisphere plays an impressive compensatory role in motor function recovery.[15 16] PMd has extensive connections to M1 as well as specific motor regions in the cerebral motor connections of the parietal and frontal lobes.[17] Sankarasubramanian et al found that the novel approach of cPMd facilitation (5 Hz rTMS) resulted in more improvement than that of cM1 inhibition (1 Hz rTMS) in severely impaired stroke patients by alleviating interhemispheric competition inflicted on weak iM1.[16] cPMd reduces inhibition of weak iM1 and provides ipsilateral access (uncrossed corticospinal and brainstem-mediated reticulospinal) to the paralysed limb to aid recovery.[15 18] Lotze et al proposed an inverse correlation between PMd activation and ipsilateral corticospinal tract (CST) integrity, verifying the potential importance of contralateral PMd for good motor performance.[19] In a study on interhemispheric cPMd-iM1 interactions, Bestmann et al used paired-coil transcranial magnetic stimulation (TMS) through concurrent TMS-functional MRI (fMRI) and confirmed that conditioning cPMd pulses had a facilitative effect on ipsilesional M1.[20] This depends on the rising excitability in ipsilesional sensorimotor areas (BA4p), which maintains direct projections to spinal motoneurons

and then increases the motor outputs of the paralysed upper limb.[20] Therefore, we expect that cPMd, as a potential therapeutic target, can facilitate the rehabilitation of upper extremity movements following ischaemic stroke.

To this end, this randomised controlled clinical trial aims to investigate whether the excitatory rTMS protocol over cPMd has a better effect than the inhibitory rTMS protocol over cM1 to compensate for the limitations of its inconsistent therapeutic effects. In addition to common clinical scales for evaluating upper limb motor function, diffusion tensor imaging (DTI) and fMRI will be used to explore changes in structural and functional connectivity, analyse the correlation between clinical function assessments and neuroimaging data and further elucidate the neurological mechanisms underlying poststroke upper limb rehabilitation.[21] Several previous investigations have shown that cM1 inhibition does not significantly increase the rehabilitation of upper extremity movements in severe stroke patients.[16 22] As a result, we hypothesise that the excitatory rTMS protocol over cPMd might be an effective therapy for the recovery of upper extremity motor performance following severe ischaemic stroke. This research may offer significant novel clues into the clinical effects of rTMS in the treatment of upper extremity impairment in ischaemic stroke patients and the underlying neuroplasticity mechanisms of TMS intervention.

## MATERIALS AND ANALYSIS

This study is a single-blind, controlled, randomised clinical trial with two groups (A and B). The study will be carried out in the Rehabilitation Medicine Centre, the First Affiliated Hospital of Nanjing Medical University, Nanjing, China. Before the trial, detailed information about the research project was provided to potential participants. After the patients voluntarily sign the informed consent form, they will be invited to participate in the trial. The improvement of adherence lies in the need for adequate communication before enrolment, so that the participants completely understand the significance and responsibilities of this study and volunteer to participate.

All eligible participants will be randomly assigned (1:1) to either group A (n=22) or group B (n=22). In consideration of research purposes, ethical perspectives, feasibility and practicality, this study was not designed with a placebo control group. Figure 1 depicts the experiment flowchart. The study protocol has been approved by the Ethics Committee of the First Affiliated Hospital of Nanjing Medical University (2020 SR-266) and registered on Chinese Clinical Trial Registry (ChiCTR2000038049, website: http://www.chictr.org.cn).

### Participants
#### Inclusion criteria
The following are the inclusion criteria: (1) ischaemic stroke confirmed by CT and/or MRI meets the diagnostic criteria for cerebral infarction in the

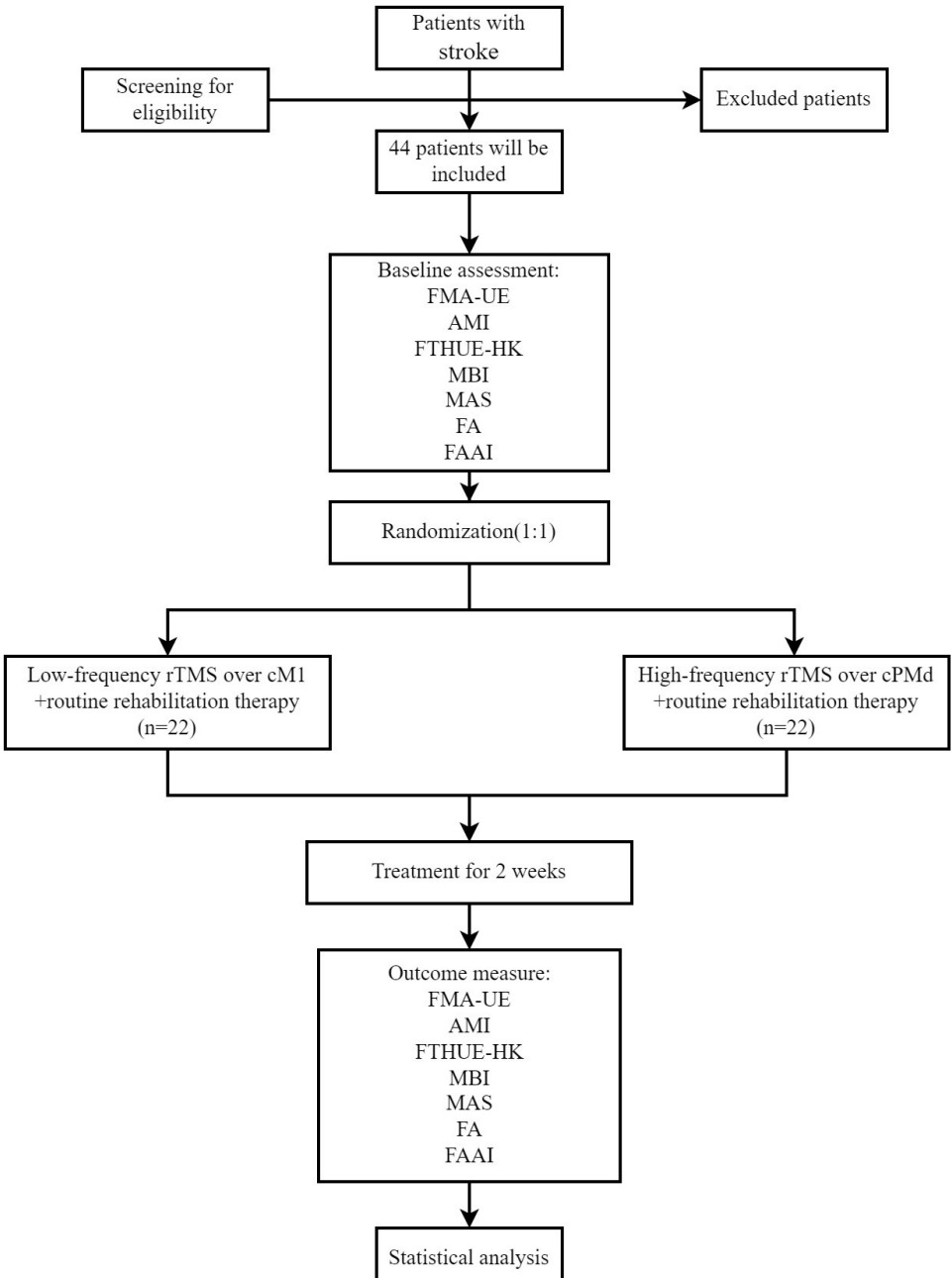

**Figure 1** Experiment flowchart. FMA-UE, Fugl-Meyer assessment of upper extremity; AMI, Arm Subscore of the Motricity Index; FTHUE-HK, Hong Kong edition of Functional Test for the Hemiplegic Upper Extremity; MBI, Modified Barthel Index; MAS, Modified Ashworth Scale; FA, fractional anisotropy; FAAI, FA Asymmetry Index; rTMS, repetitive transcranial magnetic stimulation; cM1, contralesional primary motor cortex; cPMd, contralesional dorsal premotor cortex.

cerebrovascular disease; (2) primary or unilateral onset or previous onset without residual neurological dysfunction; (3) stable vital signs and clear consciousness; (4) the age ranges from 40 to 70 years; (5) the course of disease ranges from 1 week to 3 months; (6) paralysis of the upper limbs; (7) FMA-UE score ≤22; and (8) the selected candidate or their relatives sign the informed consent form.

### Exclusion criteria
The following are the exclusion criteria: (1) epilepsy in the previous, a family history of epilepsy and taking medications that cause seizures; (2) the function of the urinary, cardiovascular, respiratory and other important systems decreases or fails; (3) severe deficits in cognition and communication that impede patient participation during evaluation and therapy; (4) posterior circulation infarction; (5) TMS and fMRI-related restrictions, including a pacemaker, an artificial metal heart valve implant, a drug treatment pump, an insulin pump, an aneurysm clip (other than titanium alloy, which is not paramagnetic) and a metal implant in the body; (6) severe cervical spondylosis includes severe cervical stenosis and

cervical spinal instability; (7) complete occlusion of the internal carotid artery; (8) direct injuries and skull defects in the stimulation area; (9) women during pregnancy; (10) severe high fever; (11) claustrophobia; and (12) inability to cooperate with fMRI examination.

### Sample size

The sample size was estimated using GPower (V.3.1.9.7). To evaluate the impact of LF-rTMS protocol over cM1 vs HF-rTMS protocol over cPMd on functional motor performance of the upper extremities evaluated utilising FMA-UE (the primary outcome) over 2 weeks, a sample size of 42 patients (21 per group) completing the intervention will be expected to reach the statistical power of 80% at the significance level of 0.05 (two-sided test), assuming a small-to-moderate effect size of Cohen's d=0.45.[23] We will randomise 44 patients (22 in each group) to account for a 5% dropout rate.

### Randomisation and blinding

After the baseline assessment is completed and the inclusion/exclusion criteria are checked, the participants will be randomised into group A or B (ratio 1:1) using SPSS software to generate random sequences. The stratification criterion is the course duration of the disease, with two strata: 1 month and 2–3 months. This study is a single-blind study. However, the evaluators who perform the measurements will be blinded to the group to which the participants belong. Special personnel are responsible for randomisation, assessment and intervention, respectively.

### Intervention

All patients will receive current routine pharmacological treatments and an 80 min rehabilitation therapy session that combines physical therapy for upper limb function, neuromuscular electrical stimulation, occupational therapy and rTMS therapy. Over the course of 2 weeks, all treatments will be administered once daily, 5 days a week (10 sessions in total). The relevant period segments of the experiment are shown in Figure 2.

rTMS will be applied with a Magneuro 100 magnetic stimulator (Vishee Medical Technology Co., Ltd., Nanjing, China) using a figure-of-eight-shaped coil for accurate targeted treatment. The centre of the coil will be tangential to the treatment plane, and the handle will be positioned at 45° from the sagittal plane. Using the surface electromyography data that will be collected from the abductor pollicis brevis on the uninjured side, the potential difference between the highest and lowest peaks will be selected to obtain the motor-evoked potential (MEP). The minimal intensity required to generate at least 5 out of 10 MEPs in a relaxed target muscle that are >50 µV is known as the resting motor threshold (RMT).[24] The group A strategy will use rTMS (1 Hz) over cM1, 90% RMT, 120 trains of 10 s duration, 2 s intertrial intervals, 1200 pulses per session and a 23 min, 58 s total duration. The group B strategy will use rTMS (10 Hz) over cPMd, 90% RMT, 80 trains of 1.5 s duration, 10 s intertrial intervals, 1200 pulses per session and a 15 min, 10 s total duration. The coil positioning will be guided throughout the

| | Study Period | | | |
|---|---|---|---|---|
| | Enrollment | Allocation | Post-allocation | |
| **TIME POINT** | W-1 | 0 | W1 | W2 |
| **ENROLLMENT:** | | | | |
| Eligibility screen | × | | | |
| Informed consent | × | | | |
| Randomization | × | | | |
| Allocation | | × | | |
| **INTERVENTIONS:** | | | | |
| LF-rTMS protocol over cM1 | | | ←——————————→ | |
| HF-rTMS protocol over cPMd | | | ←——————————→ | |
| **ASSESSMENTS:** | | | | |
| FMA-UE | × | | | × |
| AMI | × | | | × |
| FTHUE-HK | × | | | × |
| MBI | × | | | × |
| MAS | × | | | × |
| fMRI | × | | | × |

**Figure 2** Schedule of participant enrolment, interventions and assessments. LF-rTMS, low-frequency repetitive transcranial magnetic stimulation; cM1, contralesional primary motor cortex; HF-rTMS, high-frequency repetitive transcranial magnetic stimulation; cPMd, contralesional dorsal premotor cortex; FMA-UE, Fugl-Meyer assessment of upper extremity; AMI, Arm Subscore of the Motricity Index; FTHUE-HK, Hong Kong edition of Functional Test for the Hemiplegic Upper Extremity; MBI, Modified Barthel Index; MAS, Modified Ashworth Scale; fMRI, functional MRI.

experiment using a neuronavigation system (Visor2 ANT, Germany).[25] For a period of 2 weeks, each patient will receive rTMS once each day (5 days each week).

## Outcome assessment

### Outcome Assessment

#### Primary outcome

FMA-UE will be used to evaluate performance-based motor functions of the paralysed upper limbs. The FMA-UE consists of 9 major items and 33 subitems. Each item receives a rating between 0 and 3, and the total score ranges from 0 (no movement) to 66 (normal active movement). A higher score indicates lower motor impairment.[26]

#### Secondary outcomes

*Clinical and functional assessments*

AMI includes shoulder abduction, elbow flexion and pinch grip strength. The overall score of the scale is calculated by summing the scores of the three components (the highest score for each item is 33 points) and 1 point, so that a total of 100 points is reached. The score is positively correlated with the motor performance of the upper limb.[27]

FTHUE-HK includes 12 test items with seven functional levels that evaluate upper limb function as a whole. The condition in which the patient has completed the test for this level is determined according to the requirements of each level.[28] A higher level indicates that the patients have better upper extremity motor function and a greater capacity for using the upper limbs during daily activities.

MBI[29] is a 10-item rating scale that measures the quality of general life activities. The higher the score, the more independent the patients are in their daily lives. Scores >60 indicate a high likelihood of performing ADLs and maintaining a standard of living.

Modified Ashworth Scale (MAS) is the most widely used clinical scale to assess the increase in muscle tone after central nervous system lesions, which is manifested by increased resistance of joints to passive movement. According to the MAS, possible scores range from 0, 1, 1$^+$, 2, 3 to 4, with 0 reflecting normal tone and 4 reflecting fixed muscle contracture.[30]

#### DTI and fMRI indicators

Magnetic resonance (MR) scanning will be performed using a 3.0T MR scanner (Discovery 750W, GE Healthcare, USA) and a 24-channel head coil. In addition to T2-weighted image (WI) scans to exclude parenchymal brain lesions and abnormalities, anatomical images will be obtained using a 3D gradient-echo pulse sequence to collect T1WI in the sagittal plane. The following will be the parameters: repetition time (TR)=8.5 ms, time echo (TE)=2.52 ms, inversion time=450 ms, field of view (FOV)=256×256 mm², matrix size=256×256, flip angle=12°, image matrix=128×128, number of excitations (NEX)=1, bandwidth=31.25 kHz, 1.0 mm thick

slices and 192 slices. Data Processing Assistant Resting-State fMRI software (http://rfmri.org/DPARSF) will be used to preprocess the functional imaging data. DTI determines lesion size and fractional anisotropy (FA) within the posterior limb of the internal capsule (PLIC) to assess the integrity of the CST.[31] DTI is based on an echo-planar imaging sequence. The parameters will be as follows: TR=8000 ms, TE=96 ms, image matrix=112×112, FOV=22.4×22.4 cm², b=0 and 1000 s/mm², NEX=2, slice thickness=3 mm, interslice spacing=0 mm and *acceleration* factor=2. FA values and the mean FA Asymmetry Index (FAAI) are the leading indicators of tract integrity. FAAI=(FAunaffected−FAaffected)/(FAunaffected+FAaffected).[32] FAAI is in the range of −1.0 to +1.0.

### Statistical analysis

#### Clinical data analysis

Statistical analysis will be employed using SPSS statistical software (V.22.0, Chicago, IL, USA). Prior to entering data for descriptive statistics, the Shapiro-Wilk test will be used to verify normal probability. Data from the normal distribution will be given as mean and SD, but data from the non-normal distribution will be expressed as medians with IQR. Categorical variables will be described as a function of frequency as a percentage. Demographic characteristics and baseline variables will be assessed for between-group differences using statistical tests appropriate for the data type. Specifically, independent sample t-tests or non-parametric Mann-Whitney tests will be used for continuous variables, while chi-square or Fisher's exact tests will be employed for categorical variables. Repeated-measure analysis of variance will be used to estimate the therapeutic effect for primary (eg, FMA-UE) and secondary outcomes (eg, list all secondary outcomes here). Statistical significance is defined as p <0.05. The primary analysis will employ an intention-to-treat method to address non-adherence. The absence of primary outcome data will be addressed by utilising baseline data as a reference point and conducting additional sensitivity analyses. The utilisation of a randomised design precludes the use of multivariable models for confounding adjustment. Consequently, the absence of covariates will have no impact on the primary analysis.

#### Imaging data analysis

DTI datasets will be processed using the Pipeline for Analyzing Brain Diffusion Images (PANDA) toolkit (http://www.nitrc.org/projects/panda) to obtain the average FA and FAAI values of patients from the whole group.[33] The PANDA process involves three major steps: preprocessing, producing diffusion metrics and constructing networks. The bilateral PLIC, pons and precentral gyrus are selected as regions of interest. Comparisons within groups will be performed using paired t-tests and those between groups using two-sample t-tests. Structural and functional connectivity will be processed using the Infinitome software (Omniscient

Neurotechnology (2020) Infinitome). The statistical significance level is set at p <0.05.

For the Human Connectome Project Multimodal Parcellation maps, specific topic versions will be produced using a machine-learning approach. Depending on the paired functioning connection among each participant's brain maps, it will be used to model their diagnostic classification and neuropsychological examination results. A 'boosted trees' method will be applied to every case by the eXtreme Gradient Boosting (XGB) Classifier to adapt the model. This strategy offers better predictive capabilities than using a single tree.

### Data collection, management and monitoring

Data collection will be conducted by highly skilled attending physicians, therapists and radiologists who are independent of the grouping and treatment procedures. They will promptly input the gathered data into the designated case report form (CRF). Subsequently, a designated individual will digitally capture all the data while ensuring the use of unique identifiers to safeguard patient privacy and data security.

The data monitoring committee (DMC) will verify whether the conduct, generation, recording and reporting of the clinical trial comply with the regulatory requirements of the protocol, based on the monitoring plan and written standard operating procedures. By regularly reviewing the collected data, including checking the data entry and storage against predefined standards and guidelines, they examine the consistency, logic and validity of the data. They compare the entered data with the source data to confirm its accuracy and consistency. If any anomalies or potential issues are identified, the DMC will promptly investigate and correct them. They may communicate with data collectors, researchers or other relevant parties to clarify and resolve issues, ensuring the validity and reliability of the data. The goal of data monitoring is to minimise data errors and biases and ensure data quality to support accurate data analysis and reliable research conclusions.

### Harms

The intensity, frequency and other parameter settings of rTMS will comply with the safety guidelines[34] issued by the *International Transcranial Magnetic Association* in 2009. To reduce the probability of an adverse event, the participants will be rigorously screened in accordance with the exclusion and inclusion criteria. Adverse effects and events will be closely monitored during the clinical trial. If this occurs, it will be noted in the CRF. Some incidents will be monitored, and when these incidents are inappropriate, the treatment will be discontinued, such as exacerbation of the condition, serious adverse events, poor adherence leading to a loss of follow-up or the development of a new serious illness affecting the course of this protocol. This study is performed under the supervision of the Hospital Research Ethics Committee. Study progress and existing problems are semiannually reported, and adjustments are made to overcome them in a timely manner.

## ETHICS AND DISSEMINATION
### Research ethics approval

This study has been approved by the Ethics Committee of the First Affiliated Hospital of Nanjing Medical University (2020 SR-266) .

### Protocol amendments

The modifications to the protocol will be determined through in-depth discussions and careful deliberations by the Hospital Research Ethics Committee. Once the amendments receive approval from the Chinese Clinical Trial Registry, the modified protocol will be implemented for the study. The current version of the protocol is V.5.0 (date: 28 June 2021).

### Consent or assent

The researchers diligently provide comprehensive information to all eligible participants, including details about the purpose, associated risks, potential benefits and any possible adverse effects. This ensures that participants have a clear and thorough understanding of the relevant information. Only after confirming that participants have been fully informed, they are requested to sign an informed consent form, thus demonstrating their voluntary agreement to participate in the study. The improvement of adherence lies in the need for adequate communication before enrolment, so that the participants completely understand the significance and responsibilities of this study and volunteer to participate.

### Competing interests statement

The authors declare that the research was conducted in the absence of any commercial or financial relationships that could be construed as a potential conflict of interest.

### Access to data

Principal investigators and the study statistician will have access to the final dataset. To ensure confidentiality, data dispersed to project team members will be blinded of any identifying participant information.

### Dissemination policy

The findings of this study will be spread through networks of scientists, professionals and the general public as well as peer-reviewed scientific papers and presentations at pertinent conferences.

**Author affiliations**
[1]Rehabilitation Medicine Center, The First Affiliated Hospital of Nanjing Medical University, Nanjing, China
[2]School of Rehabilitation Medicine, Nanjing Medical University, Nanjing, China
[3]Department of Radiology, The First Affiliated Hospital of Nanjing Medical University, Nanjing, China

**Acknowledgements** We thank all the patients for their contributions during the completion of this study.

**Contributors** TW and YS conceived and designed the study; WD, CL, XY, YZ and LF performed the study and collected materials; HD, CG and YQ analysed the results; WD, XY and MD wrote the manuscript; YS, TW and WD helped coordinate the study and reviewed the manuscript. WD, XY, CL and HD contributed equally. All authors contributed to the article and approved the submitted version.

**Funding** This study was funded by the National Key Research & Development Program of China (2022YFC2009700), the Nanjing Municipal Science and Technology Bureau (2019060002) and the National Natural Science Fund (82302882).

**Competing interests** None declared.

**Patient and public involvement** Patients and/or the public were not involved in the design, or conduct, or reporting, or dissemination plans of this research.

**Patient consent for publication** Consent obtained from parent(s)/guardian(s).

**Ethics approval** This study has been approved by the Ethics Committee of First Affiliated Hospital of Nanjing Medical University(2020-SR-266).

**Provenance and peer review** Not commissioned; externally peer reviewed.

**ORCID iDs**
Wenjun Dai http://orcid.org/0000-0003-2625-3033
Yi Zhu http://orcid.org/0000-0003-2647-9729

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
