## [Reviewer comments · BMJ Open]

ARTICLE DETAILS

TITLE (PROVISIONAL)	Effects of repetitive transcranial magnetic stimulation over the contralesional dorsal premotor cortex on upper limb function in severe ischemic stroke: study protocol for a randomized controlled trial
AUTHORS	Dai, Wenjun; Yang, Xi; Liu, Canhuan; Ding, Hongyuan; Guo, Chuan; Zhu, Yi; Dong, Manyu; Qian, Yilun; Fang, Lu; Wang, Tong; Shen, Ying

VERSION 1 – REVIEW

REVIEWER	Tsipsios, Dimitrios Sunderland Royal Hospital
REVIEW RETURNED	17-May-2023

GENERAL COMMENTS	Excellent designed study protocol dealing with rTMS over the contralesional dorsal premotor cortex for upper limb stroke recovery. Looking forward to your study results
---

REVIEWER	Bastani, Pouya B. Johns Hopkins Medicine
REVIEW RETURNED	01-Jun-2023

GENERAL COMMENTS	The research hypothesis is novel and the protocol is clearly outlined in the paper. There are some parts of the paper that would be beyond my capability to evaluate. However, I evaluated the parts with relevance to my expertise and came to the conclusion that's outlined.
---

REVIEWER	Mehndiratta, Amit Indian Institute of Technology Delhi, Biomedical Engineering
REVIEW RETURNED	07-Jun-2023

GENERAL COMMENTS	I have reviewed the manuscript "Effects of repetitive transcranial magnetic stimulation over the contralesional dorsal premotor cortex on upper limb function in severe stroke: study protocol for a randomized controlled trial". The study design has a major flaw, group A will be treated with low-frequency (1 Hz) rTMS over the contralesional primary motor cortex (cM1), and group B will be treated with high-frequency rTMS (10 Hz) over cPMd. Both are intervention arms with different hypothesis, there is no control arm. A control arm must be part of the clinical protocol for comparative analysis with conclusion evidence. Need clarity on:
---

	What method for randomization and allocation of participant to the two intervention group will be used. Why inclusion criteria of age range 40-70 yrs, and not include adult >18 yr typically? This study is an attempt to investigate the recovery of patients with severe stroke, however, Haemorrhagic stroke has not been included. Why? All patients under both groups will be receiving physiotherapy, rTMS (HF or LF as per two groups) along with neuromuscular electrical stimulation (NES). How do authors propose to regress the rehabilitation effect observed by the neuromuscular electrical stimulation (NES) in both the groups, as rTMS and NES can have confounding effects. Why no control arm is included in the clinical protocol, either with no rTMS or sham rTMS must be included in the cohort. There should be a third group – Control group with routine standard treatment, else, impact of these in the patient cohort could play a confounding effect. Why 90%RMT is selected as the threshold for rTMS for both the groups. Is it 90% RMT or MSO? What is the total duration of rTMS therapy per day for group A and group B. A) 120 trains of 10-s duration, 2-s inter-trial intervals, and 1200 pulses per session and B) 80 trains of 1.5-s duration, 10-s inter-trial intervals, and 1200 pulses per session; doesn't appear to equal dosage. How does author justify the equivalence of these two rTMS protocol and dosage. Before statistical analysis, normality of the data must be evaluated to decide the optimal statistical test. Figure 1 must say 44 patients "will be" included instead of "were" included. Timelines for recruit, analysis and evaluation must be part of the clinical protocol paper. Line 58, Introduction: "electrical pulses to the brain" must be "magnetic pulses to the brain"? Registry name is not included in the manuscript, neither where it is registered nor the intended registry. Trial design is unclear as both are intervention groups with no control arm. The rTMS studies embarks upon the cortical excitability and sustaining the functional or the recovery in general requires the cortical excitability to be maintained, hence, a long term follow up should be included in the protocol for 1 month and 3 months. The second para in Introduction section is well written explaining the two models of the recovery, however, literature supporting the connection between PMd and M1 is only 1 (Bestmann, reference 17). More literature must be included here to support the hypothesis. Since the cortical excitability of ipsilesional side should be taken into account for recovery, taking MEP of the uninjured side is not enough, Authors must take cortical excitability of both sides. Please include the distal measures as well as the MEP is taken from APB for the final evaluation. Only shoulder elbow clinical assessments seem to be insufficient.
--	---

VERSION 1 – AUTHOR RESPONSE

Excellently designed study protocol dealing with rTMS over the contralesional dorsal premotor cortex for upper limb stroke recovery.

Looking forward to your study results

Reply: Thank you for your affirmation of the research. We will follow the experimental process with strict quality control and look forward to the later research results.

Reviewer 2

The research hypothesis is novel and the protocol is clearly outlined in the paper. There are some parts of the paper that would be beyond my capability to evaluate. However, I evaluated the parts with relevance to my expertise and came to the conclusion that's outlined.

Reply: Thank you for your recognition and valuable comments on the innovation of the experiment and the description of the scheme. We will carefully review and actively apply the improvement suggestions you have provided to ensure the enhancement of our research.

Reviewer 3

1. The study design has a major flaw, group A will be treated with low-frequency (1 Hz) rTMS over the contralesional primary motor cortex (cM1), and group B will be treated with high-frequency rTMS (10 Hz) over cPMd. Both are intervention arms with different hypothesis, there is no control arm. A control arm must be part of the clinical protocol for comparative analysis with conclusion evidence.

Reply : We appreciate your insightful feedback, particularly regarding the absence of a control arm in our study design. In this part, we aim to address your concern and provide a comprehensive explanation as to why we believe it is not necessary to include a sham group in our clinical protocol.

We understand your suggestion for including a control arm to enable comparative analysis with conclusive evidence. However, in the context of our specific study design and research objectives, we respectfully argue that the inclusion of a sham group is not essential. Allow us to provide the following justifications for our position:

Intervention Arms and Hypotheses:

We can conduct a comparative study using high-frequency rTMS over cPM to address the limitations of low-frequency rTMS over cM1 in upper limb motor function recovery in ischemic stroke. In this study, we can randomly assign ischemic stroke patients with severe damage into two groups, one receiving low-frequency rTMS over cM1 and the other receiving high-frequency rTMS over cPMd. By comparing the therapeutic effects of the two interventions, we can verify whether high-frequency rTMS over cPM is more suitable for ischemic stroke patients with severe damage.

Comparative Analysis:

While comparative analysis with a control arm is commonly employed to ascertain the specific effects of an intervention, in our study, we aim to determine the relative effectiveness of low-frequency rTMS versus high-frequency rTMS. By directly comparing the outcomes between the two intervention arms,

we can evaluate the differential effects and draw meaningful conclusions regarding the potential benefits of each intervention on motor function.

Ethical Considerations:

Given the existing evidence on the beneficial effects of rTMS in motor recovery, it would be ethically challenging to deny patients access to any form of active intervention. By focusing solely on the comparison between the two intervention arms, we can maximize the potential benefits for all participants while minimizing ethical concerns associated with the use of a sham group.

Feasibility and Practical Considerations:

Considering the logistics and resources required to conduct a clinical trial with multiple arms, including a sham group would significantly increase the complexity and sample size needed. By focusing on the comparison between the two intervention arms, we can effectively utilize our available resources and still obtain valuable insights into the differential effects of low-frequency and high-frequency rTMS.

We understand the importance of control groups in clinical research, and we appreciate your concern regarding the absence of a sham group in our study design. However, in the context of our specific research aims, the direct comparison between the low-frequency and high-frequency rTMS interventions provides us with valuable information to address our hypotheses effectively.

We have revised the manuscript to clarify the rationale behind our study design and the specific objectives we aim to achieve (line93-94, page3; line114-115, page3). We believe that the modifications we have made adequately address your concern and enhance the transparency of our research.

Once again, we sincerely appreciate your meticulous evaluation and valuable feedback. We believe that the revised manuscript now provides a comprehensive and accurate account of our study design and research objectives. We thank you for your time and consideration and look forward to your feedback on the revised version.

2.What method for randomization and allocation of participant to the two intervention group will be used.

Reply: In this study, we conducted stratified randomization based on the course of the disease, with stratification according to within one month and 2-3 months of the onset of the disease. We have revised it in line148-149, page4.

3. Why inclusion criteria of age range 40-70 yrs, and not include adult >18 yr typically?

Reply: Thank you for your review of our research. We value your suggestions and guidance. We carefully considered and weighed the selection criteria for age in our research, and ultimately determined to include adults between the ages of 40 and 70. The reasons for this selection are as follows:

Firstly, this age range has relative homogeneity, stability, and representativeness, which increases the comparability and reliability of the research results. A smaller age gradient can reduce and minimize the impact of participant age on the research results.

Secondly, the age group of 40-70 years old is the primary high-risk age range for stroke. For many research projects, selecting subjects in this age group would be more conducive to studying effective stroke intervention strategies (*Schambra 2015, Amatachaya 2016*).

4. This study is an attempt to investigate the recovery of patients with severe stroke, however, Haemorrhagic stroke has not been included. Why?

Reply: Thank you for your suggestion. The study mainly included patients with severe ischemic stroke and excluded patients with hemorrhagic stroke. We did neglect to mention this aspect in the title and text, and we have made the correction to describe the study participants as patients with severe ischemic stroke (line3, page1; line20, page1; line24, page1; line50, page2; line51, page2; line54, page2; line55, page2; line57, page2; line75, page2; line91, page3; line102, page3; line104, page3).

5. All patients under both groups will be receiving physiotherapy, rTMS (HF or LF as per two groups) along with neuromuscular electrical stimulation (NES). How do authors propose to regress the rehabilitation effect observed by the neuromuscular electrical stimulation (NES) in both the groups, as rTMS and NES can have confounding effects.

Reply: Thank you for your valuable advice. The purpose of this study is to compare whether HF rTMS over cPMd is more effective than LF rTMS over cM1 in patients with severe ischemic stroke. Routine treatments like as physiotherapy and NES are consistent and not different in the two groups. The NES control the same time, site, and frequency to avoid the influence of NES on the experimental results as much as possible. The variables in this experiment are the site and frequency of rTMS intervention, in order to study the difference in efficacy between the two intervention protocols, and no comparison is made between the efficacy of rTMS and NES.

6. Why no control arm is included in the clinical protocol, either with no rTMS or sham rTMS must be included in the cohort. There should be a third group – Control group with routine standard treatment, else, impact of these in the patient cohort could play a confounding effect.

Reply: In consideration of research purposes, ethical perspectives, feasibility, and practicality, etc., this study was not designed with a placebo control group. The specific explanation has been provided in detail in the first reply. We have revised the manuscript to clarify the rationale behind our study design and the specific objectives we aim to achieve (line93-94, page3; line114-115, page3).

7. Why 90% RMT is selected as the threshold for rTMS for both the groups. Is it 90% RMT or MSO?

Reply: In this trial, the intensity of rTMS was 90% RMT (line165, page4). According to the recent TMS guidelines (*Lefaucheur et al., 2019; Rossi et al., 2009*), there is no optimal protocol for rTMS and only a safe range of intensity (90%-130% RMT) is recommended. The 90% RMT intensity used in this study falls within the recommended safe range, and there is also some research basis for the relevant intensity to observe the efficacy of applying 90% RMT to upper limb motor function in stroke patients (*Zheng et al., 2015; Meng and Song, 2017*).

8. What is the total duration of rTMS therapy per day for group A and group B. A) 120 trains of 10-s

duration, 2-s inter-trial intervals, and 1200 pulses per session and B) 80 trains of 1.5-s duration, 10-s inter-trial intervals, and 1200 pulses per session; doesn't appear to equal dosage.

Reply: Thank you for the suggestion. We have modified in the text (line166, page4; line168, page4). The total duration of group A is 23min58s (line166, page4), while the total duration of group B is 15min10s (line168, page4). In the meantime, the total number of pulses for the two intervention schemes is the same (1200 pulses per session), therefore we consider the two intervention protocols to have an equal dosage.

9. How does author justify the equivalence of these two rTMS protocol and dosage.

Reply : Currently, research cannot effectively demonstrate the equivalence of the two intervention protocols in terms of dosage. There have been no relevant studies reporting on the differences in therapeutic effects among different combinations of stimulation parameters. In a comparative study of high-frequency (HF) rTMS and low-frequency (LF) rTMS cited in the latest TMS guidelines (*Lefaucheur et al., 2019*), patients in the high-frequency stimulation group received rTMS as follows: 3 Hz, 10 seconds, inter-train interval of 10 seconds, 40 trains, for a total of 1200 pulses on the affected hemisphere. For the low-frequency stimulation group, patients received rTMS as follows: 1 Hz, 30 seconds, inter-train interval of 2 seconds, 40 trains, for a total of 1200 pulses on the unaffected hemisphere (*Du et al., 2016b*). Based on existing literature records, this study adopted the same total number of pulses to achieve equivalence of the intervention protocols as much as possible.

10. Before statistical analysis, normality of the data must be evaluated to decide the optimal statistical test.

Reply : Before conducting statistical analysis, assessing the normality of the data is essential. Prior to entering data for descriptive statistics, the Shapiro-Wilk test will be used to verify normal probability. Data from the normal distribution will be given as mean and standard deviation, but data from the non-normal distribution will be expressed as medians with interquartile ranges. We have revised it in line216-223 ,page6.

11. Figure 1 must say 44 patients "will be" included instead of "were" included.

Reply : Thank you for your valuable suggestions, we have revised it (Figure 1).

12. Timelines for recruit, analysis and evaluation must be part of the clinical protocol paper.

Reply : Thank you for your valuable suggestion. We understand the importance of a timeline, which is why Figure 2 in this study describes the timeline for recruiting, analyzing, and assessing.

13. Line 58, Introduction: "electrical pulses to the brain" must be "magnetic pulses to the brain"?

Reply : Thank you for your suggestion. We have made modifications in the text (line59, page2).

14. Registry name is not included in in the manuscript, neither where it is registered nor the intended registry.

Reply : This is indeed our oversight, and we have made additions and modifications in the text to address it (line117-118, page3).

15 Trial design is unclear as both are intervention groups with no control arm.

Reply : Our study comprises two intervention arms, namely group A receiving low-frequency (1 Hz) rTMS over the cM1, and group B receiving high-frequency (10 Hz) rTMS over cPMd. Control other variables and eliminate other confounding factors. By comparing the therapeutic effects of the two interventions, we can verify whether high-frequency rTMS over cPMd is more suitable for ischemic stroke patients with severe damage.

In consideration of research purposes, ethical perspectives, feasibility, and practicality, etc., this study was not designed with a placebo control group. The specific explanation has been provided in detail in the first reply. We have revised the manuscript to clarify the rationale behind our study design and the specific objectives we aim to achieve (line93-94, page3; line114-115,page3).

16. The rTMS studies embarks upon the cortical excitability and sustaining the functional or the recovery in general requires the cortical excitability to be maintained, hence, a long term follow up should be included in the protocol for 1 month and 3 months.

Reply : This is indeed a limitation of our study. We only observed the immediate effects before and two weeks after the intervention to explore whether there were immediate differences in efficacy between the two intervention protocols, while ignoring their long-term effects. This is the direction of our next research. Furthermore, due to medical insurance limitations, our patients only received half a month of treatment in each course, and follow-up may not be possible in the following 1 month or 3 months.

17. The second para in Introduction section is well written explaining the two models of the recovery, however, literature supporting the connection between PMd and M1 is only 1 (Bestmann, reference 17). More literature must be included here to support the hypothesis.

Reply : We appreciate it very much for this good suggestion. We have made further descriptions accordingly to support the hypothesis (line78-85, page2-3).

18. Since the cortical excitability of ipsilesional side should be taken into account for recovery, taking MEP of the uninjured side is not enough, Authors must take cortical excitability of both sides.

Reply : Thank you very much for your valuable feedback. In this study, the amplitude of MEP on the unaffected side was solely utilized for measuring the RMT on the unaffected hemisphere. Its purpose was to determine the appropriate intensity for the TMS intervention on the unaffected hemisphere rather than serve as a metric for comparing the effects before and after the treatment. Meanwhile, the FAAI in the DTI examination of this study is the comparison of cortical excitability between the two hemispheres to understand the connectivity between the both cerebral cortex. We have made necessary further explanations in the paper (line211, page5).

19. Please include the distal measures as well as the MEP is taken from APB for the final evaluation. Only shoulder elbow clinical assessments seem to be insufficient.

Reply : Our clinical outcome measures, including FMA-UE and AMI, evaluate not only shoulder joint function but also the assessment of distal joints such as the elbow, wrist, and hand joints. Meanwhile, studies have found that for patients with severe stroke, MEPs on the affected side are mostly absent (*Kirton et al., 2010, Stinear et al., 2007*). This is due to severe damage to the corticospinal tract in the affected hemisphere, resulting in the inability of muscles on the affected side to generate normal MEP responses. In this study, we included patients with severe impairments, and most of them exhibited absent MEP responses. Therefore, MEP amplitude should be excluded as one of the outcome measures for the experiment.

VERSION 2 – REVIEW

REVIEWER	Bastani, Pouya B. Johns Hopkins Medicine
REVIEW RETURNED	09-Aug-2023
GENERAL COMMENTS	Thank you for addressing the comments diligently.

VERSION 2 – AUTHOR RESPONSE

Reviewer 1

Please can you include your explanation for the age range in the main text of your manuscript:

3. Why inclusion criteria of age range 40-70 yrs, and not include adult >18 yr typically?

Reply: We would like to extend our sincere gratitude for your meaningful comments and valuable suggestions. We have added the explanation for the age range in the main text (line56-57, page 2).

- We felt that the following comment has not been fully addressed, please can you explain why Haemorrhagic stroke was not included in this study:

4.This study is an attempt to investigate the recovery of patients with severe stroke, however, Haemorrhagic stroke has not been included. Why?

Reply: Thank you for your valuable suggestions. We carefully considered the selection criteria for type of stroke in our research, and ultimately determined to included patients with ischemic stroke and excluded patients with hemorrhagic stroke. The reasons for this selection are as follows:

Firstly, ischemic stroke and hemorrhagic stroke represent the two primary classifications of cerebrovascular accident, each underlain by distinctly divergent pathophysiological mechanisms. Ischemic stroke typically emanates from occlusion of blood flow due to atherosclerosis, whereas hemorrhagic stroke is generally the consequence of vessel rupture. Consequently, the therapeutic approaches and responses can be exhibit disparate between the two, necessitating their separate consideration in clinical research, especially when employing interventions like repetitive transcranial magnetic stimulation (rTMS).

Secondly, in the application of rTMS, the selective stimulation of specific cerebral regions is employed to modulate neuronal activity, which can yield varying outcomes. For instance, individuals with hemorrhagic stroke might exhibit divergent responses to rTMS compared to those with ischemic strokes due to the impacts of cerebral tissue damage and bleeding. This underscores the imperative to consider these two types of cerebrovascular accidents independently when designing clinical trials. Doing so is pivotal for accruing precise and reliable research findings and for assessing the efficacy and neuromechanism of rTMS with elevated accuracy.

In light of these significant pathophysiological and responsive disparities, meticulous segregation of the two stroke types in clinical trials employing rTMS is deemed indispensable for attaining more accurate evaluations of therapeutic impacts and neuromechanism. Therefore, the study mainly included patients with ischemic stroke and excluded patients with hemorrhagic stroke.